# Prevalence of depression and associated factors among pregnant women attending antenatal care in public health institutions of Awabale Woreda, East Gojjam Zone, Northwestern Ethiopia: A cross-sectional study

**Alemayehu Bantie**[1], **Getachew Mullu Kassa**[1], **Haymanot Zeleke**[1], **Liknaw Bewket Zeleke**[2], **Bewket Yeserah Aynalem**[1]*

**1** Health Science College, Debre Markos University, Debre Markos, Ethiopia, **2** School of Women's and Children's Health, University of New South Wales Sydney, Sydney, Australia

* by123bewket@gmail.com

**Data Availability Statement:** All relevant data are within the manuscript. Authors are unable to share

## Abstract

### Background

Antenatal depression is a serious health problem and has negative consequences for the mother, fetus, and the entire family. However, it is a neglected component of care especially bay health care providers for women in pregnancy. The purpose of this study was to assess the prevalence of depression and associated factors among pregnant women attending antenatal clinics in public health institutions, in the Awabale Woreda.

### Method

An institutional-based cross-sectional study was conducted in 2018 and a stratified sampling technique was used to select the study health institutions. All seven public health institutions in Awabale District were included to select 393 mothers and the sample size was proportionally allocated based on the number of target mothers. We used EpiData version 3.1software for data entry and SPSS version 20 software for cleaning and analysis. A Bivariable logistic regression analysis was used to identify the association between each outcome variable and the factor. Again, a multivariable logistic regression analysis was employed to identify factors associated with each outcome variable, and variables with a p-value less than 0.05 were taken as significant variables. Edinburgh Postnatal Depression Scale was used to declare the presence of antenatal depression with a cut point score of 13 and above.

### Result

This study showed that 63(17.8%) pregnant mothers had antenatal depressive symptoms. Women who were employed 85% reduced to develop antenatal depression than

additional data due to indentifying information and ethical restrictions. For data access requests, please reach out to Bekalu Kassie (PG and Data Ethics Committee coordinator) at email: bekalukassiedmu@gmail.com.

**Funding:** The authors received no specific funding for this work.

**Competing interests:** The authors declare that they have no competing interests.

**Abbreviations:** AOR, Adjusted Odds Ratio; CI, Confidence Interval; COR, Crude Odds Ratio; EPDS, Edinburgh Postnatal Dépression Scale.

housewives [AOR = 0.15(0.001–0.25)]. Pregnant women who attended high school and above educational level were 18 times more likely to develop antenatal depression than women who had no formal education [AOR18.15 (2.73–120.76)]. Women who had poor husband feeling on the current pregnancy were 4.94 more likely to develop antenatal depression than women who had good partner feeling on the current pregnancy [AOR = 4.94(95%CI: 1.78–13.72)]. Women who had a history of depression were 8.2 times to develop antenatal depression than women who had no history of depression [AOR = 8.22 (95%CI: 2.87–23.57)].

## Conclusion

This study revealed that approximately one-fifth of pregnant women developed antenatal depression. Women's occupational status, educational status, previous history of depression, and poor husband feeling on the current pregnancy were the significant factors of antenatal depression.

## Introduction

Pregnancy is the period from the fertilization of the egg by sperm to the delivery of the fetus and usually lasts 40 weeks. Starting from the last normal menstrual period, it is divided into three trimesters, each lasting three months [1]. It is a distinctive social and biological event in a woman's life [2, 3].

A depressive disorder is an illness that involves the body, mood, and thoughts in which the person has unrelenting sentiments of unhappiness and irrelevance and a lack of aspiration to engage in previously enjoyable activities that last more than two weeks. When these sentiments last for a short period, it may be a case of feeling sadness [4]. It is the most frequent expressive disorder in women and the general population, and one in every five people has depression with more than the twofold increased incidence in women than men [2]. Psychiatric history, stressful life events, lack of social support, teenage pregnancy, low educational level, low income, and violence against women are the factors that make depression prevalent in pregnant women [3, 5, 6].

Antenatal depression means depression that starts from conception to delivery [7]. Pregnant women experience symptoms similar to general depression and may interfere with their normal day-to-day activities. It may occur at any stage of pregnancy and can be a reaction to the pregnancy itself, due to health issues, major life stresses, genetic and biochemical basis, or due to a continuation or relapse of a pre-pregnancy condition [8]. Antenatal depression ends with poor fetal, infant development, and maternal outcomes like small for gestational age, prematurity, intrauterine growth problem; postnatal depression preeclampsia, anemia; educational problem, malnutrition, respiratory disorders, and mental retardation [9–13].

There are studies, which examined the prevalence and associated factors of antenatal depression in low-income countries like Ethiopia; but the result of the studies was not consistent [14, 15]. Some studies have a limitation on addressing pregnant mothers in all age groups and excluded illiterate mothers from the study [3, 16].

Even though the Federal Democratic Republic of Ethiopia's National Mental Health Strategy promotes a decentralized approach in which mental health services are to be offered from local health institutions up to tertiary hospitals, the service is not being given as expected [17,

18]. Thus the purpose of this study was to assess the prevalence of depression and associated factors among pregnant women attending antenatal clinics in public health institutions, in Awabale Woreda.

## Methods

### Study area and period

This study was done in Awabale woreda, Amhara region, Northwest Ethiopia in 2018. Awabale woreda is the capital city of East Gojjam Zone in Amhara regional state, Northwest Ethiopia, which is located at 300 km from Addis Ababa, the capital city of Ethiopia, and 265 km from Bihar Dar, the capital city of Amhara region. Awabale woreda has a 146260 total population; about 37413 are females aged 15–49 years [19]. There are seven public health institutions eligible for this study in Awabale Woreda and Antenatal care is provided in all public health institutions. The Woreda has an annual ANC plan of 5590(1309 in Lumame health center, 648 in Shebela health center, 641 in Wejele health center, 665 in Lega health center, 901 in Yesenbet health center, 765 in Tsid-Maryam health center and 661 in Lumame primary hospital) [19].

### Study design

An institutional-based cross-sectional study design was employed.

### Study participants

All pregnant women attending antenatal care in Awabale Woreda public health institutions were the source population. All pregnant women attending antenatal care in selected public health institutions during the study period were the study population. Pregnant women at any age of gestation who come to the selected health institutions for antenatal care visits during the study period were included in the study and we excluded those who were critically ill during the data collection period.

### Sample size

The sample size was calculated based on a single population proportion formula assumption. The prevalence of Antenatal depression was 31.2% from a study conducted in Adama, Ethiopia [15], and a 5% margin of error was used.

$initial sample size = \left(Z\frac{a}{2}\right)^2 * \frac{p(1-p)}{w2} = 1.96^2 * \frac{0.312(1-0.312)}{(0.05)^2} = 330$. We have added a non-response rate of 10% and 330*0.10 = 33. Then the final sample size was 330+33 = 363.

### Sampling technique and procedure

Stratified sampling was undertaken to select the study of health institutions. All seven public health institutions in Awabale District were included and the sample size was proportionally allocated based on the number of target mothers (Lumame Health Center = 85, Shebila Health Center = 42, Wejele Health Center = 42, Lega Health Center = 43, Yesenbet Health Center = 58, Tsid-Maryam Health Center = 50, Lumame Primary Hospital = 43). All eligible and consenting women attending for antenatal care during the study period were taken into the study consecutively until the sample size was reached.

### Study variables

**Dependent variable.**   Depression during pregnancy (present/absent).

### Independent variables

**Socio-demographic factors.**    Maternal age, educational status, marital status, occupation of the mother and family, monthly income.

**Obstetric factors.**    Gravidity, gestational age, history of abortion, unplanned pregnancy.

**Psycho-social factors.**    Social support, relationship with a partner, husband support, violence.

**Clinical factors.**    A self-history of depression, family history of depression, history of chronic illness.

### Operational definitions

**Antenatal depression.**    It is an illness in which pregnant women have an Edinburgh Post-natal Depression Scale (EPDS) score of 13 and above [20, 21].

**Social support.**    Is support from the community ranges from a score of 3 to 14 according to the OSLO social support scale. A score of 3–8 is poor support, 9–11 is moderate support, and 12–14 is strong support.

### Data collection procedure and data quality control

To assure the data quality, data were collected with face-to-face interviews by attained BSc Midwife in each institution after one-day data collection training was given to them together with two MSc holder midwife supervisors. The questionnaire was structured and pre-tested which was first prepared in English and translated to local (Amharic) language and then again translated back to English by language experts. A pretest was conducted on 18 pregnant women of the sample size in Debre-Elias woreda health institution and the necessary correction on the tool was employed accordingly.

### Data collection and measurement tool

EPDS was administered to detect symptoms of depression and their socio-demographic data along with obstetric and psychosocial factors. EPDS is a 10 item questionnaire scored from zero up to three (higher score indicating more depressive symptoms), that has been validated for detecting depression in antepartum and postpartum samples in many countries. The instrument was validated in public health centers in Addis Ababa for postpartum use and showed a sensitivity of 84.6% and specificity of 77.0% at the cutoff score of 7/8 [21]. Those pregnant women who score 13 and above were categorized as depressed women while pregnant women who scored below 13 were considered as non-depressed women [20, 21].

The OSLO-3 item social support scale was used to measure social support for pregnant women. Partners feeling on the current pregnancy can be defined as the sensation of pregnant women about the feeling of their partners concerning the current pregnancy. It was measured by asking whether her partner feeling on the current pregnancy was good or poor. Similarly, husband/partner support was assessed by asking women emotions about their partner's support to the health of the fetus and continuation of pregnancy. A structured Amharic version questioner containing socio-demographic characteristics, obstetric history, and psychosocial history, history of clinical factors, history of violence, and history of substance abuse was administered.

### Data analysis technique

EpiData version 3.1 software was used for data entry and SPSS version 20 was used for analysis. Bivariate logistic regression was employed to identify an association between independent and

dependent variables. Variables having a P-value of less than 0.2 in the bivariate logistic regression analysis were fitted into the multivariable logistic regression model to manage confounders. The 95% confidence interval of odds ratio was computed and a variable having P-value less than 0.05 in the multivariable logistic regression analysis was considered as statistically significant.

## Ethical consideration

The study was approved by the Institutional Health Research Ethics Review Committee of Debre Markos University, College of health science. An official letter was written from Debre Markos University to the selected health institutions. The participants enrolled in the study were informed about the study objectives, expected outcomes, benefits, and the risks associated with it. Written consent was taken from the participants before the interview.

## Result

### Socio-demographic factors

Three hundred fifty-four study participants gave the response to the questionnaire, giving a response rate of 97.5. %. The majorities of the respondents were Amhara 351 (99.2%) and orthodox by religion were 293 (82.8%). Three hundred thirty-five (94.6%) of the women were married and 150(44.8%) were farmers in their occupation (Table 1).

**Table 1. Socio-demographic characteristics of pregnant women attending ANC in Awabale district, East Gojjam zone, Northwest Ethiopia 2018(n = 354).**

| Variables | | Frequency | Percentage |
|---|---|---|---|
| Maternal Age | $\leq 19$ | 17 | 4.8 |
| | 20–34 | 262 | 74.0 |
| | $\geq 35$ | 75 | 21.2 |
| Religion | Orthodox Christian | 293 | 82.8 |
| | Muslim | 61 | 17.2 |
| Ethnicity | Amhara | 351 | 99.2 |
| | Oromo | 3 | 0.8 |
| Marital status | Not married | 19 | 5.4 |
| | Married | 335 | 94.6 |
| Educational status of the respondent | Illiterate(Unable to read and write) | 170 | 48.0 |
| | Elementary school (grade 1–8) | 91 | 25.7 |
| | High school and above | 93 | 26.3 |
| Occupation of the respondent | Employed | 69 | 19.5 |
| | Running personal business | 60 | 16.9 |
| | Housewife | 225 | 63.6 |
| Educational status of the husband | Illiterate(Unable to read and write) | 187 | 55.8 |
| | Elementary school (grade 1–8) | 57 | 17.0 |
| | High school and above | 91 | 27.2 |
| Occupation of the husband | Employed | 77 | 23.0 |
| | Running personal business | 108 | 32.2 |
| | Farmer | 150 | 44.8 |
| Residence | Rural | 189 | 53.4 |
| | Urban | 165 | 46.6 |
| Monthly income | $\leq 500$ | 45 | 12.7 |
| | >500 | 309 | 87.3 |

**Table 2. Obstetrics and clinical characteristics of pregnant women attending ANC in Awabale district, East Gojjam zone, Northwest Ethiopia 2018(n = 354).**

| Variables | | Frequency | Percentage |
|---|---|---|---|
| History of previous pregnancy | No | 145 | 41.0 |
| | Yes | 209 | 59.0 |
| Number of pregnancies | Primigravidas | 145 | 41.0 |
| | Multigravida | 209 | 59.0 |
| Previous history of abortion | N0 | 167 | 79.9 |
| | Yes | 42 | 20.1 |
| Type of abortion | Spontaneous | 37 | 88.1 |
| | Induced | 5 | 11.9 |
| History of stillbirth | No | 186 | 89.0 |
| | Yes | 23 | 11.0 |
| Is the current pregnancy planned | No | 48 | 13.6 |
| | Yes | 306 | 86.4 |
| Trimester | First trimester | 78 | 22.0 |
| | Second trimester | 195 | 55.1 |
| | Third trimester | 81 | 22.9 |
| Gestational age at first ANC | Before 16 weeks | 254 | 71.8 |
| | After 16 weeks | 100 | 28.2 |
| Previous history of depression | No | 296 | 83.6 |
| | Yes | 58 | 16.4 |
| Family history of a depressive episode | No | 345 | 97.5 |
| | Yes | 9 | 2.5 |
| History of chronic illness | No | 316 | 89.3 |
| | Yes | 38 | 10.7 |

## Obstetric and clinical characteristics

Two hundred nine (59%) of the respondents were gravida one and above. Around 58(16.4%) and 9 (2.5%) of the respondents had a previous history of depression and a family history of depression respectively and 38(10.7%) had a history of chronic illness (Table 2).

## Psychosocial, history of violence and substance use

More than Three-fourths (77.4%) of the respondents reported their husbands are happy with the occurrence of current pregnancy and a half (50.8%) of women explored good baby father support to the health of the fetus and continuation of pregnancy. Most 33(47.1%) and 249 (70.3%) of the participants complained health-related problems were the most frequent emotionally disturbing factor in their lifetime and during the current pregnancy respectively. According to the OSLO social support scale, 112(31.6%) women have poor, 160(45.2%) moderate, and 82(23.2%) strong social support. Seventy-three (20.6%) of women's had a history of violence in their lifetime (Table 3).

## Prevalence of antenatal depression

About 63 (17.8%) of respondents had antenatal depressive symptoms (EPDS score ≥13). More than half (57%) of the respondents were able to laugh and see the funny side of things. On the other hand, (58.5%) of pregnant women felt sad or miserable most of the time (Table 4).

**Table 3. Psychosocial, history of violence and substance use of pregnant women attending ANC in Awabale district, East Gojjam zone, Northwest Ethiopia 2018 (n = 354).**

| Variables | | Frequency | Percentage% |
|---|---|---|---|
| Husband feeling on the current pregnancy | Happy(Good) | 274 | 77.4 |
| | Not happy(Poor) | 80 | 22.6 |
| Husband support to the health of the fetus and continuation of pregnancy | Very good | 148 | 41.8 |
| | Good | 180 | 50.8 |
| | Not good | 26 | 7.3 |
| Social support(Based on OSLO social support scale) | Poor | 112 | 31.6 |
| | Moderate | 160 | 45.2 |
| | Strong | 80 | 23.2 |
| The emotionally disturbing factor for the last twelve month | No | 284 | 80.2 |
| | Yes | 70 | 19.8 |
| Psychosocial problems addressed at the antenatal clinic | No | 29 | 8.2 |
| | Yes | 325 | 91.8 |
| Enough information about the pregnancy and expected labor at ANC | No | 23 | 6.5 |
| | Yes | 331 | 93.5 |
| History of violence | No | 281 | 79.4 |
| | Yes | 73 | 20.6 |
| Lifetime history of substance use | No | 196 | 55.4 |
| | Yes | 158 | 44.6 |
| History of substance use during the current pregnancy | No | 200 | 56.5 |
| | Yes | 154 | 43.5 |

## Factors associated with antenatal depression

Educational status of the respondent [COR = 1.68(95% CI: 1.02, 3.45)], occupation of respondents [COR = 0.37 (95% CI: 0.15, 0.91)], occupation of partners [COR = 0.57(5% CI: 0.27, 1.2)], trimester [COR = 2.4(95% CI: 1.30, 4.42)], unplanned pregnancy [COR = 6.85(95% CI: 3.55,13.22)], poor husband support and feelings [COR = 8.21(95% CI: 4.52, 4.91)], history of depression [COR = 4.21(95% CI: 2.26, 7.84)], history of violence [COR = 4.79(95% CI: 2.65, 8.64)], history of substance use in previous pregnancy [COR = 3.02(95% CI: 1.70, 5.36)] and history of substance use in current pregnancy [COR = 2.70(95% CI: 1.54, 4.75)] were significant factors in the bivariate regression analysis (Table 5).

In the multivariable regression analysis; occupational status of the respondent [AOR = 0.15 (95%CI 0.001–0.25)], women educational status of high school and above[AOR = 16.23(95% CI 2.46–107.27)], poor husband feeling on the current pregnancy[AOR = 4.86(95%CI: 1.74–13.58)] and the previous history of depression [AOR = 7.26(95%CI: 2.52–20.93)] were significant factors for antenatal depression (Table 5).

## Discussion

This study aimed to assess the prevalence of depression and associated factors among pregnant women attending antenatal care in public health institutions of Awabale Woreda, East Gojam zone, Amhara National Regional State, Northwestern Ethiopia, during March-April 2018. The study finding of antenatal depression (17.8%) in the current study was in line with similar reports in Bangladesh (18%) and Ethiopia (19.9%) [22, 23]. The finding of this research was lower than the studies done in China (28.5%) [7], 29.9% and 31.2% in Ethiopia [20, 24], 33.8 in Tanzania [16], 38.5% and 47% in South Africa [16, 25]. This difference might be due to the

**Table 4. EPDS measurement of antenatal depression among pregnant women attending ANC in Awabale district, East Gojjam zone, Northwest Ethiopia 2018 (n = 354).**

| Variables | | Frequency | Percentage |
|---|---|---|---|
| I have been able to laugh and see the funny side of things: | As much as I always could | 203 | 57.3 |
| | Not quite as much now | 110 | 31.1 |
| | Not so much now | 24 | 6.8 |
| | Not at all | 17 | 4.8 |
| I have look forward with enjoyment to things: | As much as I ever did | 204 | 57.6 |
| | Rather less than I used to | 99 | 28.0 |
| | Less than I used to | 28 | 7.9 |
| | Hardly at all | 23 | 6.5 |
| I have blamed myself unnecessarily when things went wrong: | Yes, most of the time | 111 | 31.4 |
| | Yes, some of the time | 103 | 29.1 |
| | Not very often | 128 | 36.2 |
| | No, never | 12 | 3.4 |
| I have been anxious or worried for no good reason: | No, not at all | 133 | 37.6 |
| | Hardly ever | 82 | 23.2 |
| | Yes, sometimes | 125 | 35.3 |
| | Yes, very often | 14 | 4.0 |
| I have felt scared or panicky for no very good reason: | Yes, quite a lot | 88 | 24.9 |
| | Yes, sometimes | 112 | 31.6 |
| | No, not much | 118 | 33.3 |
| | No, not at all | 36 | 10.2 |
| Things have been getting on top of me: | Yes, most of the time I haven't been able to cope at all | 105 | 29.7 |
| | Yes, sometimes I haven't been coping as well as usual | 135 | 38.1 |
| | No, most of the time I have coped quite well | 108 | 30.5 |
| | No, I have been coping | 6 | 1.7 |
| I have been so unhappy that I have had difficulty sleeping: | Yes, most of the time | 139 | 39.3 |
| | Yes, sometimes | 118 | 33.3 |
| | Not very often | 68 | 19.2 |
| | No, not at all | 29 | 8.2 |
| I have felt sad or miserable: | Yes, most of the time | 207 | 58.5 |
| | Yes, quite often | 103 | 29.1 |
| | Not very often | 31 | 8.8 |
| | No, not at all | 13 | 3.7 |
| I have so unhappy that I have been crying | Yes, most of the time | 282 | 79.7 |
| | Yes quite often | 53 | 15 |
| | Only occasionally | 18 | 5.1 |
| | No, never | 1 | 0.3 |
| The thought of harming myself has occurred to me: | Yes, quite often | 0 | 0 |
| | Sometimes | 340 | 96.0 |
| | Hardly ever | 12 | 3.4 |
| | Never | 2 | 0.6 |

difference in their population demographic characteristics, study design, period, and the difference in their investigatory or diagnostic tools.

Prevalence of antenatal depression was associated with the occupation of the women, educational status, history of depression, and poor husband feeling in the current pregnancy. Women who were running employees were 85% reduced to develop antenatal depression than

**Table 5. Factors associated with antenatal depression among pregnant women attending ANC in Awabale district, East Gojjam zone, Northwest Ethiopia 2018 (n = 354).**

| variables | | Depression | | COR(95%CI) | AOR(95%CI) |
|---|---|---|---|---|---|
| | | No | Yes | | |
| Occupational status of the respondent | Employed | 63 | 6 | 0.37 (0.15, 0.91) | 0.15(0.001, 0.25) |
| | Running personal business | 49 | 11 | 0.87(0.42, 1.81) | 0.48(0.07, 3.21) |
| | Housewife | 179 | 46 | 1 | 1 |
| Educational status of the respondent | Illiterate | 147 | 23 | 1 | 1 |
| | Elementary school | 64 | 27 | 1.68(1.02, 3.45) | 1.08(0.18, 6.4) |
| | High school and above | 80 | 13 | 1.04(0.5, 2.16) | 16.23(2.46, 107.27) |
| Occupational status of the husband | Employed | 66 | 11 | 0.57(0.27, 1.2) | 1.47(0.21, 10.04) |
| | Running personal business | 102 | 6 | 0.20(0.81, 0.5) | 0.22(0.044, 1.09) |
| | Farmer | 116 | 34 | 1 | 1 |
| Trimester | First | 88 | 10 | 2.4(1.30, 4.42) | 1.90(0.47, 7.74) |
| | Second | 154 | 41 | 1 | 1 |
| | Third | 69 | 12 | 0.65(0.32, 1.32) | 0.87(0.28, 3.69) |
| Planned pregnancy | No | 24 | 24 | 6.85(3.55,13.22) | 3.06(0.73, 12.87) |
| | Yes | 267 | 39 | 1 | 1 |
| Social support | Poor | 81 | 31 | 2.40(1.30, 4.42) | 1.14(0.28, 4.61) |
| | Moderate | 138 | 22 | 1 | 1 |
| | Strong | 72 | 10 | 0.87(0.39, 1.94) | 0.92(0.24, 3.61) |
| Husband feeling on the current pregnancy | Good | 248 | 26 | 1 | 1 |
| | Poor | 43 | 37 | 8.21(4.52, 4.91) | 4.86(1.74, 13.58) |
| Previous history of depression | No | 256 | 40 | 1 | 1 |
| | Yes | 35 | 23 | 4.21(2.26, 7.84) | 7.26(2.52, 20.93) |
| History of violence | No | 247 | 34 | 1 | 1 |
| | Yes | 44 | 29 | 4.79(2.65, 8.64) | 1.27(0.34, 4.79) |
| Lifetime history of substance use | No | 116 | 42 | 1 | 1 |
| | Yes | 175 | 21 | 3.02(1.70, 5.36) | 1.88(0.73, 4.84) |
| History of substance use in the current pregnancy | No | 114 | 40 | 1 | 1 |
| | Yes | 177 | 23 | 2.70(1.54, 4.75) | 0.09(0.004, 1.82) |

housewives [AOR = 0.15(0.001–0.25)]. This might be due to that, those who are employees may have social relationships and may have the satisfaction that makes the women economically independent [26]. Also, those housewife women are expending most of their time at home and alone. This loneliness may put them in depression [27].

Women with high school and above educational levels were 18 times more likely to develop antenatal depression than women who had no formal education [16.23(95%CI 2.46–107.27)]. This might be due to their difficulty in managing interpersonal relationship strains related to academic performance pressure and inability to translate their additional education into better mental health outcomes [28]. On the other hand, different studies in different countries report the association of lower educational status with an increased prevalence of antenatal depression [22, 26, 29].

Those women who had a history of depression had 7.26 times the odds of developing antenatal depression than women who had no history of depression [AOR = 7.26(95%CI: 2.52–20.93)]. The comparable association was also reported from studies conducted in developing and developed countries [20]. This might be due to physical and hormonal changes occurring during pregnancy and the recurrence of depressive symptoms [26]. Conversely, the personal

history of previous psychiatric illness was not found to be a significant risk factor for antenatal depression in a study conducted in Lahore, Pakistan [30].

Pregnant women who had poor husband feeling on the current pregnancy were 4.86 times more likely to develop antenatal depression as compared with good husband feeling towards current pregnancy [AOR = 4.86(95%CI: 1.74–13.58)]. This is possible because those partners who had good feelings about the pregnancy authorize the women on their home responsibilities and help women to have care for their health and the health of the fetus. It might be also due to the effect of a poor husband's feeling on diminishing partner support [20].

The finding of this study shows no significant association between partner occupational status, trimester, unplanned pregnancy, social support, history of violence, and substance use in the multivariable model. This result seems consistent with other findings [20, 31]. On the other hand, contrary to this finding, those women who had a history of substance use had a higher risk of developing antenatal depression [16].

## Limitations of the study

We cannot be certain that individuals with an EPDS score $\geq$ of 13 had depressive illness without confirmation. Social desirability bias due to face-to-face interviews and using cross-sectional studies which do not show causality is also the limitation of this study.

## Conclusion

The prevalence of antenatal depression in women attending antenatal care services at public health institutions is high. Women's occupational statuses, educational status, previous history of depression, and poor husband feeling on the current pregnancy were significant factors for antenatal depression.

## Supporting information

**S1 Questionnaire.**
(DOCX)

## Acknowledgments

We would like to express our deepest heartfelt thanks to Debre Markos University for their permission to do this research and we gratefully acknowledge all study individuals for their participation in the study.

## Author Contributions

**Conceptualization:** Alemayehu Bantie, Getachew Mullu Kassa, Haymanot Zeleke, Liknaw Bewket Zeleke, Bewket Yeserah Aynalem.

**Data curation:** Alemayehu Bantie, Getachew Mullu Kassa, Haymanot Zeleke, Liknaw Bewket Zeleke, Bewket Yeserah Aynalem.

**Formal analysis:** Alemayehu Bantie, Getachew Mullu Kassa, Haymanot Zeleke, Liknaw Bewket Zeleke, Bewket Yeserah Aynalem.

**Funding acquisition:** Alemayehu Bantie, Getachew Mullu Kassa, Haymanot Zeleke, Liknaw Bewket Zeleke, Bewket Yeserah Aynalem.

**Investigation:** Alemayehu Bantie, Getachew Mullu Kassa, Haymanot Zeleke, Liknaw Bewket Zeleke, Bewket Yeserah Aynalem.

**Methodology:** Alemayehu Bantie, Getachew Mullu Kassa, Haymanot Zeleke, Liknaw Bewket Zeleke, Bewket Yeserah Aynalem.

**Project administration:** Alemayehu Bantie, Getachew Mullu Kassa, Haymanot Zeleke, Liknaw Bewket Zeleke, Bewket Yeserah Aynalem.

**Resources:** Alemayehu Bantie, Getachew Mullu Kassa, Haymanot Zeleke, Liknaw Bewket Zeleke, Bewket Yeserah Aynalem.

**Software:** Alemayehu Bantie, Getachew Mullu Kassa, Haymanot Zeleke, Liknaw Bewket Zeleke, Bewket Yeserah Aynalem.

**Supervision:** Alemayehu Bantie, Getachew Mullu Kassa, Haymanot Zeleke, Liknaw Bewket Zeleke, Bewket Yeserah Aynalem.

**Validation:** Alemayehu Bantie, Getachew Mullu Kassa, Haymanot Zeleke, Liknaw Bewket Zeleke, Bewket Yeserah Aynalem.

**Visualization:** Alemayehu Bantie, Getachew Mullu Kassa, Haymanot Zeleke, Liknaw Bewket Zeleke, Bewket Yeserah Aynalem.

**Writing – original draft:** Alemayehu Bantie, Getachew Mullu Kassa, Haymanot Zeleke, Liknaw Bewket Zeleke, Bewket Yeserah Aynalem.

**Writing – review & editing:** Alemayehu Bantie, Getachew Mullu Kassa, Haymanot Zeleke, Liknaw Bewket Zeleke, Bewket Yeserah Aynalem.

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
