## [Decision Letter · Decision Letter 0]

15 Jul 2021

PONE-D-21-01660

Prevalence and Associated Factors of Pregnant Women attending Antenatal Care in Public Health Institutions of Awabale Woreda, Northwestern Ethiopia: a cross-sectional study

PLOS ONE

Dear Dr. Aynalem,

Thank you for submitting your manuscript to PLOS ONE. After careful consideration, we feel that it has merit but does not fully meet PLOS ONE’s publication criteria as it currently stands. Therefore, we invite you to submit a revised version of the manuscript that addresses the points raised during the review process.

The manuscript has been evaluated by three reviewers, and their comments are available below.

The reviewers have raised a number of major concerns. They particularly note the need for greater detail and clarity in the manuscript’s reporting of the data collection and analyses. They specifically request further detail on the development and validation of the study instruments and further refinement in the statistical modelling. In addition, they note that greater depth of discussion is required with regards to prior literature, in both the introduction and discussion. They also noted the need for professional language assistance, which may help to address the aforementioned issues.  

We look forward to receiving your revised manuscript.

Kind regards,

Avanti Dey, PhD

Staff Editor

PLOS ONE

Journal Requirements:

Reviewers' comments:

Reviewer's Responses to Questions

**Comments to the Author**

1. Is the manuscript technically sound, and do the data support the conclusions?

Reviewer #1: Partly

Reviewer #2: No

Reviewer #3: No

2. Has the statistical analysis been performed appropriately and rigorously? 

Reviewer #1: No

Reviewer #2: No

Reviewer #3: No

3. Have the authors made all data underlying the findings in their manuscript fully available?

Reviewer #1: No

Reviewer #2: Yes

Reviewer #3: Yes

4. Is the manuscript presented in an intelligible fashion and written in standard English?

Reviewer #1: Yes

Reviewer #2: No

Reviewer #3: No

5. Review Comments to the Author

Reviewer #1: Thank you very much for your interest in antenatal depression.

1. While the study appears to be sound and a very important study area, the authors have, however, failed to address how their findings relate to previous research in this area. A lot of work has been done especially in Ethiopia regarding antenatal depression to the extent that there are three recent systematic literature reviews [Ayano et al (2019), Getinet et al (2018), Zegeye et al (2018)]. The published work such as by Hanlon et al and Bitew et al has laid the foundation in this area of study. In the southern region of Africa, work by Honikman et al, Ng'oma et al, Stewart et al and Kaiyo et al also highlight the magnitude of the burden of antenatal depression. Could the authors align their introduction and discussion to reference such existing literature? The introduction should also highlight why the authors think that antenatal depression is a "serious health problem", and they should also bring out the known "negative consequences for the mother, fetus and the entire family" so as to convince the reader that surely there is need to tackle antenatal depression.

2. Kindly state your eligibility criteria that you used during sampling procedure and clearly explain why it was important to include those women and exclude the ones that were not eligible. The authors need to clearly explain the type of health institutions that are in their study area and why they had to use stratified sampling method to sample the health institutions. How many health facilities are in this region and how many did they use as study sites? How did they sample their participants from each study health facility? The study sample size was 363 but, in the end, they interviewed 354 participants. Does that mean there were no more eligible participants at the study sites? Considering this was a cross-sectional study, I am not sure why they could not reach the required minimum number of participants.

3. To save on repeating themselves, the authors could just have combined STUDY VARIABLES, OPERATIONAL DEFINITIONS & MEASUREMENT TOOLS into one subheading eg STUDY MEASURES and define their terms. For example:

Independent Variable

Antenatal depression, defined as an illness in which pregnant women have an EPDS score of 13 and above, was assessed using the EPDS... then go on to describe the tool.

Same applies to the other variables.

4. Were the study instruments validated for the specific population? The authors mention that the EPDS was validated in Addis Ababa and at a cut-off score of 7/8, it had a sensitivity of 84.6% and specificity of 77.0%. But they went on to use a cut off score of 13 to define their depression caseness. Can the authors please justify their cut off score? Please explain more about the OSLO-3 item social support scale. The authors should also explain the data collection procedures simply and clearly so that the reader can be able to picture what they did exactly. This also helps with reproducibility of their work. As it is, it is not clear how they collected their data.

5. The authors' statistical analyses are very vague. While they have explained how the associations between antenatal depression and the independent variables was identified, they did not explain how their study sample would be described. Descriptive statistics would be very important to describe the sample. The results show that frequencies (and percentages) were computed but this was not stated in their data analysis technique. What were the confounding variables that the authors considered in their multivariate analysis? And how did they come up with the confounders?

6. In the results section, kindly label the tables clearly, and clearly reference your tables in the text. Do not just put tables that have no link to the text. The authors should also explain their results for the reader to clearly follow their arguments. This is particularly about the section headed "Factors associated with antenatal depression"; the following sentence "Age, marital status.... .... ... were significant factors in the bivariate regression analysis". it would be helpful to put in the odd ratios, 95%CI and p values next to the factors eh "Age [COR (95%CI), p value], marital status [COR (95%CI), p value], ... ... ..were significant factors". It would be helpful also to mention the direction of the association. For example, Pregnant women aged below 16 were X times more likely to develop antenatal depression or whatever your results show. Please insert the table that shows the results of the bivariate logistic regression analysis.

7. Like mentioned before, please align your discussion to the previous research done especially in sub-Saharan region. There is also more recent work on this subject you could cite. The authors have highlighted their weaknesses, there is need to show that the study had some strengths well. Otherwise, why would it be worthwhile. Another weakness they can highlight is their study design - cross sectional studies do not show causality and also the social desirable bias due to face to face interviews they administered. With these results, what do the authors recommend to policy makers, researchers, and practitioners. What is their take home message?

8. The study is very important, but the language is unclear. The sentences are too long, winding and thus difficult to understand. I advise the authors to work with a copy editor to improve the flow and readability of the manuscript.

Reviewer #2: Thank you authors for conducting this study. It is timely and important. Here are my comments

1. Revise the title including the topic of study

2. Considering the sample size, high volume of published information on the topic so far in Ethiopia using the dichotomous EDPDS, and most importantly the nature of your outcome variable which is latent variable, I strongly recommend making re-analysis using structural equation modeling and re-write the manuscript in such way. You can use stress-process model or other theoretical model to conceptualize the problem.

Reviewer #3: Dear the authors, thank you for presenting a critical public health issue in the developing countries. This study may be important for the local health planners and researchers.

However, there are plenty of studies in the topic area in the world as well as in Ethiopia. Your paper do not bring any new evidence to the scientific world. I have doubt the relevance of your study considering enough evidence of your topic title in your country. The paper lacks strong justification of the reasons of conducting the study topic in the area

The variables under your study were limited(important variables were missed in your study) and it is difficult to reach conclusion without including all the relevant confounding variables.

From the scientific perspective your study lacks many qualities; e.g. you have large confidence interval which this may be due to inadequate sample size, Data itself is not consistent, or you have outliers in the data, or you have poorly specified model, or you have (partial) collinearity between the variables.

There are a lot of inconsistencies in your study data presentation. e.g in the abstract section you have mentioned women who were running their business were 0.015 times less likely of developing antenatal depression than housewives [AOR=0.015(0.001-0.236)]. however in result section Women who were running their business were 85% reduced to develop antenatal depression than housewives [AOR=0.15(0.001-0.25)]. Moreover, what is found in the abstract, result section are completely different from what is found in the tables. Please see the regression analysis table and description carefully and revise.

more specifically

your study topic lacks the outcome under the study: please revise

Abstract section:

if you add sample size, the study population, analysis run to identify the associated factors and the cut off score

Large confidence interval

Introduction section

The gap for studying this title is not adequately mentioned

There are abundant studies in Ethiopia. this study did not bring any new evidence

the severity, the impact in the study area was not mentioned well

Method section

You have stated stratified sampling as your sampling techniques, so how is all pregnant woman during the study period included in your study?

you have taken the proportion from the study done in Adama. Is Adama and Awabale woreda has the same characteristics in terms of population, health care services, etc.... ?

you have used consecutive sampling(non-probability): I would rather use random sampling(e.g systematic sampling) for generalizability issue. If you have used stratified sampling why did you use consecutive sampling at the end?

what is the reliability score of your tool in your study?

You have mentioned the tool was validated in Ethiopia on postpartum depression. what about antepartum use?

your study was on antenatal depression

How didi you measure the fitness of the model?

how did you measure the strength of association?

Result section

be consistent in the description.

Discussion section

The justification for the difference in the prevalence is not based on your findings, realistic and scientific.

Even you have used two Ethiopian studies for comparison. one is higher and another is lower than your finding. this needs justification

most of the reasons for justification are shallow

Limitations of the study

The EPDS measurement tool is a measurement scale, not a diagnostic tool.....This can't be the limitation. as your objective is not to diagnose the patient

Reference

References were not written to the journal standard

Tables

What is your base to classify rural - urban

Monthly income: the categorization of income seems arbitrary

Only 5 respondents had induced abortion . How did you fit type of abortion variable (the cell count 5) in your regression analysis?

what is the question asked to measure family history of a depressive episode

how does the participants knows depressive episode?

What is the question asked to measure History of chronic illness

how does the participants knows chronic illness? what are the types?

6. PLOS authors have the option to publish the peer review history of their article (what does this mean?). If published, this will include your full peer review and any attached files.

Reviewer #1: **Yes: **Malinda Kaiyo-Utete

Reviewer #2: **Yes: **Abel Dadi

Reviewer #3: No

---

## [Author Response · Author response to Decision Letter 0]

17 Jul 2021

thank you for your constructive comments

---

## [Decision Letter · Decision Letter 1]

31 Aug 2021

PONE-D-21-01660R1

Prevalence of Depression and Associated Factors among Pregnant Women Attending Antenatal Care in Public Health Institutions of Awabale Woreda, East Gojjam Zone, Northwestern Ethiopia: a cross-sectional study

PLOS ONE

Dear Dr. Aynalem,

Thank you for submitting your manuscript to PLOS ONE. After careful consideration, we feel that it has merit but does not fully meet PLOS ONE’s publication criteria as it currently stands. Therefore, we invite you to submit a revised version of the manuscript that addresses the points raised during the review process.

We look forward to receiving your revised manuscript.

Kind regards,

Jianguo Wang, PhD

Academic Editor

PLOS ONE

Journal Requirements:

Additional Editor Comments (if provided):

Please carefully address the comments and improve the quality of this manuscript. Most of all is to clearly hightlight the importance or novelty of this research.

Reviewers' comments:

Reviewer's Responses to Questions

**Comments to the Author**

1. If the authors have adequately addressed your comments raised in a previous round of review and you feel that this manuscript is now acceptable for publication, you may indicate that here to bypass the “Comments to the Author” section, enter your conflict of interest statement in the “Confidential to Editor” section, and submit your "Accept" recommendation.

Reviewer #1: (No Response)

Reviewer #2: (No Response)

Reviewer #3: (No Response)

2. Is the manuscript technically sound, and do the data support the conclusions?

Reviewer #1: Partly

Reviewer #2: (No Response)

Reviewer #3: No

3. Has the statistical analysis been performed appropriately and rigorously? 

Reviewer #1: No

Reviewer #2: (No Response)

Reviewer #3: No

4. Have the authors made all data underlying the findings in their manuscript fully available?

Reviewer #1: No

Reviewer #2: (No Response)

Reviewer #3: Yes

5. Is the manuscript presented in an intelligible fashion and written in standard English?

Reviewer #1: Yes

Reviewer #2: (No Response)

Reviewer #3: No

6. Review Comments to the Author

Reviewer #1: This has improved very much from the previous version. However, the authors still have not addressed the issue of the importance of their work. What have they done differently considering that a lot of work has already been done in Ethiopia. Could they please provide a sound justification for their work?

The methodology section still needs to be revised. There is a lot of repetition.

I suggest the authors send their work to a professional editor.

Reviewer #2: My comments have not been well addressed. There has been a lot of studies on this area so far and I do not think this study brought new evidence that has not been known.

Reviewer #3: Dear authors thank you for presenting important public health problem.

I have forwarded comments and questions in the first review. I don't see much improvement from the first version.

Please go through each comments and questions one by one and respond appropriately.

Please find attached the comments and questions.

Thank you

7. PLOS authors have the option to publish the peer review history of their article (what does this mean?). If published, this will include your full peer review and any attached files.

Reviewer #1: **Yes: **Malinda Kaiyo-Utete

Reviewer #2: **Yes: **Abel Dadi

Reviewer #3: No

---

## [Author Response · Author response to Decision Letter 1]

1 Sep 2021

Thank you for all of your constructive comments

---

## [Decision Letter · Decision Letter 2]

30 Sep 2021

PONE-D-21-01660R2Prevalence of Depression and Associated Factors among Pregnant Women Attending Antenatal Care in Public Health Institutions of Awabale Woreda, East Gojjam Zone, Northwestern Ethiopia: a cross-sectional studyPLOS ONE

Dear Dr. Aynalem,

Thank you for submitting your manuscript to PLOS ONE. After careful consideration, we feel that it has merit but does not fully meet PLOS ONE’s publication criteria as it currently stands. Therefore, we invite you to submit a revised version of the manuscript that addresses the points raised during the review process.

We look forward to receiving your revised manuscript.

Kind regards,

Jianguo Wang, PhD

Academic Editor

PLOS ONE

Additional Editor Comments (if provided):

Please carefully address each comment raised by reviewers and improve the quality of manuscript.

Reviewers' comments:

Reviewer's Responses to Questions

**Comments to the Author**

1. If the authors have adequately addressed your comments raised in a previous round of review and you feel that this manuscript is now acceptable for publication, you may indicate that here to bypass the “Comments to the Author” section, enter your conflict of interest statement in the “Confidential to Editor” section, and submit your "Accept" recommendation.

Reviewer #1: (No Response)

Reviewer #3: (No Response)

2. Is the manuscript technically sound, and do the data support the conclusions?

Reviewer #1: Partly

Reviewer #3: Partly

3. Has the statistical analysis been performed appropriately and rigorously? 

Reviewer #1: Yes

Reviewer #3: No

4. Have the authors made all data underlying the findings in their manuscript fully available?

Reviewer #1: Yes

Reviewer #3: Yes

5. Is the manuscript presented in an intelligible fashion and written in standard English?

Reviewer #1: No

Reviewer #3: Yes

6. Review Comments to the Author

Reviewer #1: The authors have not addressed some of the issues raised in the previous review. For example, they state that there is paucity of data on antenatal depression in Ethiopia, yet there is a lot of work published from 2006 (Hanlon et al) to 2021 - a list of these articles have been provided in the attachment.

Their participants were aged between 15-49 years of age. Participants below 18 would ethically require an assent and parental consent to participate in research. However, in their manuscript, the authors do not show how this was obtained. Could they have considered such women as emancipated adults. If so, they should provide reference.

There are so many grammatical and spelling errors. Could the authors consider proof reading (maybe use of a professional editor) to correct these.

Their methodological section is not flowing smoothly. They have repeated a lot of things unnecessarily.

Their referencing style should be consistent with the journal's requirements.

Please find attached comments to the article

Reviewer #3: Dear authors,

Though there are changes from the previous draft, most of my comments and questions are not addressed well.

Please go through each questions and comments and respond

7. PLOS authors have the option to publish the peer review history of their article (what does this mean?). If published, this will include your full peer review and any attached files.

Reviewer #1: No

Reviewer #3: No

---

## [Author Response · Author response to Decision Letter 2]

1 Oct 2021

thank you for your constructive comments and suggestions. you teach me a lot

---

## [Decision Letter · Decision Letter 3]

29 Nov 2021

PONE-D-21-01660R3Prevalence of Depression and Associated Factors among Pregnant Women Attending Antenatal Care in Public Health Institutions of Awabale Woreda, East Gojjam Zone, Northwestern Ethiopia: a cross-sectional studyPLOS ONE

Dear Dr. Aynalem,

Thank you for submitting your manuscript to PLOS ONE. After careful consideration, we feel that it has merit but does not fully meet PLOS ONE’s publication criteria as it currently stands. Therefore, we invite you to submit a revised version of the manuscript that addresses the points raised during the review process.

Please insert comments here and delete this placeholder text when finished. Be sure to:This is the third-round review.Please pay your attentions to the comments from reviewers and significantly improve your manuscript. ==============================

We look forward to receiving your revised manuscript.

Kind regards,

Jianguo Wang, PhD

Academic Editor

PLOS ONE

Reviewers' comments:

Reviewer's Responses to Questions

**Comments to the Author**

1. If the authors have adequately addressed your comments raised in a previous round of review and you feel that this manuscript is now acceptable for publication, you may indicate that here to bypass the “Comments to the Author” section, enter your conflict of interest statement in the “Confidential to Editor” section, and submit your "Accept" recommendation.

Reviewer #3: (No Response)

2. Is the manuscript technically sound, and do the data support the conclusions?

Reviewer #3: Partly

3. Has the statistical analysis been performed appropriately and rigorously? 

Reviewer #3: Yes

4. Have the authors made all data underlying the findings in their manuscript fully available?

Reviewer #3: Yes

5. Is the manuscript presented in an intelligible fashion and written in standard English?

Reviewer #3: No

6. Review Comments to the Author

Reviewer #3: Dear authors, thank you for revising the manuscript. There are some changes from the previous drafts.

kindly find some of the remining comments.

7. PLOS authors have the option to publish the peer review history of their article (what does this mean?). If published, this will include your full peer review and any attached files.

Reviewer #3: No

---

## [Author Response · Author response to Decision Letter 3]

1 Dec 2021

Dear reviewer #3 thank you for your constructive comments and suggestions. Here are some responses to raised issues.

1. I still has concern on wide confidence interval. the authors should explain this "

Like other variables, [AOR18.15 (2.73-120.76)] is gained from the analysis process of the data. This may be due to different reasons like small sample size or inappropriate responses of the study participants 

2. Significant percentages of your study participants had low levels of educational status. so, how could they know whether their previous diagnosis was depression or other mental health problems. this is ambiguous unless the medical record of the study respondents reviewed 

They got the information from their health care providers who diagnosed them.

3. if all pregnant women in the study period included in your study, you have used consecutive sampling technique which is not a probability sampling technique. in that sense it is difficult to generalize your finding. I expect randomization for generalize 

We were talking about the study population here. But we can use consecutive sampling techniques in special cases like busy work areas.

4. I really don't see the importance of this. 

The Federal Democratic Republic of Ethiopia's National Mental Health Strategy promotes a decentralized approach in mental health services from local health institutions up to tertiary hospitals but service is not being given as expected. So this is important to get brief information about the prevalence and factors of depression during pregnancy. 

5. Please put the reliability of EPDS and Oslo social support scales in your study (cronbatch alpha result) 

This tool is a validated tool so putting the reliability of the validated tool may not have a value.

6. better to mention some variables were assessed using yes or no response which might not reflect the.....

Since we asked yes/no questions, we are not sure they were giving true (diagnosed responses), so we cannot mention the variables that were not confirmed

---

## [Decision Letter · Decision Letter 4]

4 Feb 2022

PONE-D-21-01660R4Prevalence of Depression and Associated Factors among Pregnant Women Attending Antenatal Care in Public Health Institutions of Awabale Woreda, East Gojjam Zone, Northwestern Ethiopia: a cross-sectional studyPLOS ONE

Dear Dr. Aynalem,

Thank you for submitting your manuscript to PLOS ONE. After careful consideration, we feel that it has merit but does not fully meet PLOS ONE’s publication criteria as it currently stands. Therefore, we invite you to submit a revised version of the manuscript that addresses the points raised during the review process.

This is the last chance for your revision.

I hope that efforts should be made to improve the quality of this manuscript. 

We look forward to receiving your revised manuscript.

Kind regards,

Jianguo Wang, PhD

Academic Editor

PLOS ONE

Reviewers' comments:

Reviewer's Responses to Questions

**Comments to the Author**

1. If the authors have adequately addressed your comments raised in a previous round of review and you feel that this manuscript is now acceptable for publication, you may indicate that here to bypass the “Comments to the Author” section, enter your conflict of interest statement in the “Confidential to Editor” section, and submit your "Accept" recommendation.

Reviewer #1: (No Response)

2. Is the manuscript technically sound, and do the data support the conclusions?

Reviewer #1: Partly

3. Has the statistical analysis been performed appropriately and rigorously? 

Reviewer #1: No

4. Have the authors made all data underlying the findings in their manuscript fully available?

Reviewer #1: No

5. Is the manuscript presented in an intelligible fashion and written in standard English?

Reviewer #1: No

6. Review Comments to the Author

Reviewer #1: Review of the Manuscript: PONE-D-21-01660_R4

“Prevalence of Depression and Associated Factors among Pregnant Women Attending Antenatal Care in Public Health Institutions of Awabale Woreda, East Gojjam Zone, Northwestern Ethiopia: a cross-sectional study”

The manuscript reads much better compared to the previous versions. However, I still have some issues with the manuscript.

Introduction

The authors are still to address the issue of the rationale for their study. As I mentioned previously, a lot of work on antenatal depression has been done in Ethiopia since 2009 (Hanlon et al., 2009; Hanlon et al., 2010). Recently, three systematic literature reviews have been published summarizing the burden of antenatal depression in Ethiopia (Dadi et al., 2020; Gertinet Ayano, 2019; Zegeye et al., 2018). So, please explain what your study is adding to the body of knowledge.

The authors have stated that “The majority of studies on the prevalence of antenatal depression and associated factors have been conducted in developed countries”. However, there has been a lot of work that has been done in developing countries, particular in African countries (M. Kaiyo-Utete & T. Magwali, 2020; MacGinty et al., 2020; Redinger, 2018; Stewart et al., 2014; van der Westhuizen et al., 2018; van Heyningen et al., 2018; Weobong et al., 2014) to mention but a few. Please refer to such studies for your introduction to be more convincing.

The authors mention that “There are studies undertaken on antenatal depression” as a limitation of the studies that were done in low-income countries. Their study is on antenatal depression so how can this be a limitation. Can they please revise their limitations?

Methods and Results

Can the authors please detail how they came up with their sample size?

EPDS is a screening tool hence the pregnant women will have depressive symptoms, not depression.

The authors should revise their statement “About 63 (17.8%) had antenatal depression.”

What do the authors mean by “Two hundred nine (59%) of respondents had a history of pregnancy”? Is it that they were multipara? Please use the correct medical terminology.

The authors could summarise the results of bivariate regression in a table and show the odds ratios (95%:CI; p value) to show the variables that were associated with antenatal depression. The narration “Age, marital status, educational status of the respondent, occupation of respondents and partners, history of abortion and stillbirth, trimester, unplanned pregnancy week at first ANC, poor husband support and feelings, social support, emotionally disturbing factors, history of depression, history of violence and substance use were significant factors in the bivariate regression analysis” is bare of the statistics.

7. PLOS authors have the option to publish the peer review history of their article (what does this mean?). If published, this will include your full peer review and any attached files.

Reviewer #1: No

---

## [Author Response · Author response to Decision Letter 4]

4 Feb 2022

Dear reviewer#1 and editors I would like to forward my heart felt gratitude for your comments and time you spent on this paper. Here are point by point responses for your questions you raised.

 Please explain what your study is adding to the body of knowledge.

 The studies you mentioned as they were done in Ethiopia were done before 11 years so that there may be possible population and environmental change. Again even though there are some current studies in Ethiopia, no study done in study done in the study area which is different in economic and socio-cultural aspect. 

 Please refer to such studies for your introduction to be more convincing

 Corrected 

 Can they please revise their limitations?

 Corrected 

 Can the authors please detail how they came up with their sample size?

 Corrected 

 The sample size was calculated based on a single population proportion formula assumption. The prevalence of Antenatal depression was 31.2 % from a study conducted in in Adama, Ethiopia [15] and a 5% margin of error was used. initialsamplesize =〖(Z a/2)〗^2*(p(1-p))/w2=〖1.96〗^2*0.312(1-0.312)/((〖0.05)〗^2 )=330. We have added a non-response rate of 10% and 330*0.10=33. Then the final sample size was 330+33=363.

 EPDS is a screening tool hence the pregnant women will have depressive symptoms, not depression.

The authors should revise their statement “About 63 (17.8%) had antenatal depression.”

 Corrected 

 What do the authors mean by “Two hundred nine (59%) of respondents had a history of pregnancy”? Is it that they were multipara? Please use the correct medical terminology.

 Corrected 

 Bare of the statistics 

 Corrected

---

## [Decision Letter · Decision Letter 5]

4 Apr 2022

PONE-D-21-01660R5Prevalence of Depression and Associated Factors among Pregnant Women Attending Antenatal Care in Public Health Institutions of Awabale Woreda, East Gojjam Zone, Northwestern Ethiopia: a cross-sectional studyPLOS ONE

Dear Dr. Aynalem,

Thank you for submitting your manuscript to PLOS ONE. After careful consideration, we feel that it has merit but does not fully meet PLOS ONE’s publication criteria as it currently stands. Therefore, we invite you to submit a revised version of the manuscript that addresses the points raised during the review process.

We look forward to receiving your revised manuscript.

Kind regards,

Jianguo Wang, PhD

Academic Editor

PLOS ONE

Journal Requirements:

Reviewers' comments:

Reviewer's Responses to Questions

**Comments to the Author**

1. If the authors have adequately addressed your comments raised in a previous round of review and you feel that this manuscript is now acceptable for publication, you may indicate that here to bypass the “Comments to the Author” section, enter your conflict of interest statement in the “Confidential to Editor” section, and submit your "Accept" recommendation.

Reviewer #1: All comments have been addressed

2. Is the manuscript technically sound, and do the data support the conclusions?

Reviewer #1: Partly

3. Has the statistical analysis been performed appropriately and rigorously? 

Reviewer #1: Yes

4. Have the authors made all data underlying the findings in their manuscript fully available?

Reviewer #1: No

5. Is the manuscript presented in an intelligible fashion and written in standard English?

Reviewer #1: Yes

6. Review Comments to the Author

Reviewer #1: The authors can still reference the vast work that has been done in Ethiopia already. I appreciate that their work is in a different economical region compared to the previous work but this gives us a better understanding of the burden of antenatal depression in their country. I would have wanted to hear more on what is different from these other studies in their discussion.

7. PLOS authors have the option to publish the peer review history of their article (what does this mean?). If published, this will include your full peer review and any attached files.

Reviewer #1: **Yes: **Malinda Kaiyo-Utete

---

## [Author Response · Author response to Decision Letter 5]

5 Apr 2022

Dear reviewers, thank you for your time you spent on this manuscript and your constructive comments you provided. Here below, we have provided response to your question. 

1. We have checked the references and we got an error on reference number 8 and 19.

2. Based on your suggestion we corrected the reference number 8, but the reference number 19 is the annual report of the district which is not published. so, it will be better if you use as a direct reference.

---

## [Editor Report · Decision Letter 6]

11 Jul 2022

Prevalence of Depression and Associated Factors among Pregnant Women Attending Antenatal Care in Public Health Institutions of Awabale Woreda, East Gojjam Zone, Northwestern Ethiopia: a cross-sectional study

PONE-D-21-01660R6

Dear Dr. Aynalem,

We’re pleased to inform you that your manuscript has been judged scientifically suitable for publication and will be formally accepted for publication once it meets all outstanding technical requirements.

Kind regards,

Jianguo Wang, PhD

Academic Editor

PLOS ONE

Additional Editor Comments (optional):

Please check your English and presentation before final submission. 'Pregnant women had high school and above educational level 18 times higher odds of developing antenatal depression than women who had no formal education' is a little confusing statement.
---

## [Editor Report · Acceptance letter]

5 Oct 2022

PONE-D-21-01660R6 

Prevalence of Depression and Associated Factors among Pregnant Women Attending Antenatal Care in Public Health Institutions of Awabale Woreda, East Gojjam Zone, Northwestern Ethiopia: a cross-sectional study 

Dear Dr. Aynalem:

I'm pleased to inform you that your manuscript has been deemed suitable for publication in PLOS ONE. Congratulations! Your manuscript is now with our production department. 

Kind regards, 

on behalf of

Dr. Jianguo Wang 

Academic Editor

PLOS ONE